# Anomalous isotope effect on mechanical properties of single atomic layer Boron Nitride

Alexey Falin[1], Haifeng Lv[2], Eli Janzen[3], James H. Edgar [3], Rui Zhang [4], Dong Qian[4], Hwo-Shuenn Sheu [5], Qiran Cai [1], Wei Gan[1], Xiaojun Wu [2], Elton J. G. Santos[6,7] & Lu Hua Li [1] ✉

The ideal mechanical properties and behaviors of materials without the influence of defects are of great fundamental and engineering significance but considered inaccessible. Here, we use single-atom-thin isotopically pure hexagonal boron nitride ($h$BN) to demonstrate that two-dimensional (2D) materials offer us close-to ideal experimental platforms to study intrinsic mechanical phenomena. The highly delicate isotope effect on the mechanical properties of monolayer $h$BN is directly measured by indentation: lighter $^{10}$B gives rise to higher elasticity and strength than heavier $^{11}$B. This anomalous isotope effect establishes that the intrinsic mechanical properties without the effect of defects could be measured, and the so-called ultrafine and normally neglected isotopic perturbation in nuclear charge distribution sometimes plays a more critical role than the isotopic mass effect in the mechanical and other physical properties of materials.

The ideal mechanical properties and behaviors of crystals are of fundamental and engineering importance, but bulk and surface defects inevitably make the measurement of ideal mechanical performance impossible. The emergence of new materials, such as whiskers[1] and single-crystal metals[2] provided us with some low defect level crystals. Recently, two-dimensional (2D) materials, such as graphene[3] and atomically thin boron nitride (BN)[4] and transition metal dichalcogenides[5,6] showed outstanding mechanical properties due to less disorder in small numbers of atoms, no surface defects due to their 2D nature, and simple chemical composition. Furthermore, they could be free of defects in the effective measuring area that is normally less than 20 nm in radius in atomic force microscope (AFM) based indentation method. Therefore, 2D materials might offer us "playgrounds" to experimentally study highly subtle mechanical phenomena without

the effect of defects for discovering new knowledge not possible before.

An isotope effect on material's elasticity and strength is one of such subtle phenomena. Although neutron numbers slightly affect the nucleus size, charge distribution, and electronic wavefunction of an atom, the isotopic mass effect is typically the most significant in light weight elements. That is, isotopic mass changes the zero-point energy, vibrational energy levels, lattice anharmonicity of interatomic interactions, which in turn influences lattice parameters, elastic constants, and bonding strength[7]. More specifically, a lighter isotope normally has higher vibrational energy, longer bond length, and weaker bond strength than those of a heavier isotope at the lowest energy level[8], but lighter isotopes acquire comparatively less energy as the temperature increases, which might reverse the above-mentioned effects at

[1]Institute for Frontier Materials, Deakin University, Geelong Waurn Ponds Campus, Waurn Ponds, Geelong, VIC 3216, Australia. [2]Hefei National Laboratory for Physical Sciences at the Microscale, School of Chemistry and Material Sciences, CAS Key Laboratory of Materials for Energy Conversion and CAS Center for Excellence in Nanoscience, University of Science and Technology of China, Hefei, Anhui 230026, China. [3]Tim Taylor Department of Chemical Engineering, Kansas State University, Manhattan, KS 66506, USA. [4]Department of Mechanical Engineering, The University of Texas at Dallas, Richardson, TX 75080, USA. [5]National Synchrotron Radiation Research Center, Hsinchu 300, Taiwan. [6]Institute for Condensed Matter Physics and Complex Systems, School of Physics and Astronomy, The University of Edinburgh, Edinburgh EH9 3FD, UK. [7]Higgs Centre for Theoretical Physics, The University of Edinburgh, Edinburgh EH9 3FD, UK. ✉e-mail: luhua.li@deakin.edu.au

elevated temperatures[9,10], especially in the case of light elements such as hydrogen and deuterium[11]. Other isotope effects such as the isotopic nuclear size and charge effects are deemed ultrafine and hard to observe.

Experimentally measuring the isotope effect on the mechanical properties of crystals has been challenging. Haussuhl and Skorczyk[12] first attempted this using a sound-velocity technique in 1969, and the elastic constant of $^7$LiD was 1.5% larger than that of $^7$LiH. In 1993, Ramdas et al.[13] introduced another method, i.e., Brillouin scattering to show that the elastic moduli of isotopically enriched $^{13}$C diamond was slightly higher than that of $^{Nat}$C diamond. However, a later Brillouin scattering study produced contradictory conclusions[14]. Furthermore, the elastic constants of the same isotopic materials obtained from the Brillouin scattering were quite different from those measured by the sound-velocity methods. The isotope effect on the mechanical properties could not be detected using traditional stress-strain measurements despite the availability of many sophisticated experimental techniques and a large variety of isotopically pure crystals nowadays.

Here, we use one-atom-thin $h$BN with different boron isotope concentrations to demonstrate that 2D materials can be used as model platforms to discover new phenomena by studying their intrinsic mechanical properties without the effect of defects. The highly subtle isotope effect on the mechanical properties is directly measured by nanoindentation: the Young's modulus ($E$) decreases with increased neutron number, i.e., $E$ for $^{10}$BN > $^{Nat}$BN > $^{11}$BN, and the mean values of its fracture strength have the same trend. Although justified by ab initio density functional theory (DFT) at both 0 K and room temperature and Synchrotron-based X-ray diffraction (XRD), these results contradictory to the commonly considered isotopic mass effect are attributed to the ultrafine isotopic nuclear charge effect that could be present in other elements close to B in the periodic table.

## Results and Discussion
### Materials and characterization
Single atomic layers of $h$BN with different boron isotope contents were chosen in this study for three reasons. First, the isotope effect in BN can be detected at room temperature, as its Debye temperature ($T_\theta$) is ~400 K[15]. Second, the isotope effect on lattice constants is normally more prominent on light elements because of the mass effect, i.e. diatom's oscillation frequency is inversely proportional to the square root of the mass of the two atoms. The two stable isotopes of B, i.e., $^{10}$B and $^{11}$B, have a relatively large mass difference of ~10%. Third, isotopically pure $h$BN single crystals are available[7,16] for mechanical exfoliation of high-quality atomically thin sheets[4,17].

The $^{10}$BN and $^{11}$BN bulk crystals were grown by the nickel-chromium solvent method at atmospheric pressure,[16] and the single crystal domains without defects were a few tens of microns, as indicated by a previous study involving crystals prepared by the same method at similar time[18]. According to our previous study, they had 99.2% $^{10}$B and 99.9% $^{11}$B, respectively[17]; naturally occurring N containing > 99.6% $^{14}$N and < 0.4% $^{15}$N was deemed isotopically pure. According to the previous study, the atomically thin $^{10}$BN and $^{11}$BN sheets were mechanically exfoliated on silicon wafers covered by 90 nm silicon oxide (SiO$_2$/Si) with pre-patterned arrays of micro-wells with radii of 350–800 nm. For comparison purposes, atomically thin naturally occurring boron nitride ($^{Nat}$BN) sheets were exfoliated under the same conditions using single crystals produced by the high-pressure Ba–BN solvent method[19]. Figure 1a shows the optical microscopy images of a 1 L $^{11}$BN covering ten micro-wells. The corresponding AFM image obtained in a repulsive tapping mode shows a thickness of 0.59 nm (Fig. 1b, c). Examples of 1 L $^{10}$BN and $^{Nat}$BN as well as the cross-section of the suspended region of the 1 L $^{11}$BN can be found in Supplementary Figs. 1, 2. The transmission electron microscopy (TEM) study of the suspended atomically thin isotopic BN was reported in our previous publication[17].

The chemical composition, structure, and purity of the three crystals were probed by Synchrotron-based near-edge X-ray absorption fine structure (NEXAFS) spectroscopy. Figure 1d shows that their NEXAFS spectra in the B K-edge region were identical and not dependent on the isotope. That is, all three crystals showed sharp π* resonances at 192.0 eV and broad σ* resonances at higher energies, corresponding to the transitions of the B 1$s$ core electrons to the unoccupied anti-bonding π* and σ* orbitals with B − 3 N bonding character following the dipole selection rule, respectively. The results affirmed the high crystallinity of the samples, and the lack of satellite peaks indicated their high chemical purity.

The different isotope compositions were manifested by Raman spectroscopy. Figure 1e compares the typical Raman $E_{2g}$ peaks of the suspended 1 L $^{10}$BN, $^{Nat}$BN, and $^{11}$BN along with their bulk crystals (at ~1/50 intensity). Similar to $^{Nat}$BN, the suspended 1 L $^{10}$BN and $^{11}$BN showed similar Raman frequencies and full width at half maximums (FWHMs) to those of their bulk crystals due to the absence of strain caused by substrates[20]. The average Raman frequencies of $^{10}$BN, $^{Nat}$BN, and $^{11}$BN were 1393.2 ± 0.3 (number of measurements $N = 17$), 1367.5 ± 0.4 ($N = 17$) and 1358.4 ± 0.3 ($N = 17$) cm$^{-1}$, respectively (Fig. 1e). These values roughly followed a linear relation to the atomic mass change indicated by the dashed line in Fig. 1f[21], where the small deviation of 1 L $^{Nat}$BN was due to static isotopic mass disorder[22,23].

### Mechanical tests by indentation
The mechanical properties of the 1 L $^{10}$BN, $^{Nat}$BN, and $^{11}$BN were measured by AFM-based nanoindentation[4]. More details of the method are provided in the Supplementary Note 1. Because the single crystal domains without defect were a few tens of microns[18] and our effective measuring radius was less than 20 nm, the chance to encounter defects during the indentation was small. Load-displacement curves were obtained from the centers of the suspended BN sheets using cantilevers with diamond tips. The curves were then fitted by[24]:

$$f = f_0 + k_1(\delta - \delta_0) + k_2(\delta - \delta_0)^3, \qquad (1)$$

where $k_1 = \sigma_0^{2D}\pi$; $k_2 = E^{2D}\left(\frac{q^3}{a^2}\right)$; $a$ is the radius of the suspended sheet; $\sigma_0^{2D}$ is the effective pre-tension of the sheets; $E^{2D}$ is the effective Young's modulus; $q = 1/(1.049 - 0.15v - 0.16v^2)$ is a dimensionless coefficient related to Poisson's ratio ($v$); $f_O$ and $\delta_O$ are starting (zero) point coordinates in the force ($f$) - displacement ($\delta$) curves. The radius was determined by AFM, and the inaccuracy had no effect on the Young's modulus and only a negligible effect on fracture strength. During the fitting process, $f_O$, $\delta_O$, $k_1$, and $k_2$ were taken as variables. In comparison to the work by Lee et al.[3], the displacement of the membrane ($Z$) and applied load ($F$) during indentation are replaced by $F = f - f_0$, and $Z = \delta - \delta_0$, respectively. This relationship better fits the linear and cubic parts of the load-displacement curve, and it also helps to identify the coordinates of the starting (zero) point of indentation, improving the accuracy of the deduced mechanical properties. For all BN sheets, the same effective thickness (0.334 nm) and Poisson ratio (0.211) were used.

The typical load-displacement curves of the 1 L $^{10}$BN, $^{Nat}$BN, and $^{11}$BN as well as the corresponding fittings are compared in Fig. 2a. All curves were fitted up to the cubic regime to meet the pre-requirement of the transformed Schwerin (cubic) term[25] in Eq. (1) to obtain the correct Young's modulus[26]. The cubic regimes were estimated by the load-cubed displacement ratio ($F/\delta^3$) versus membrane displacement ($\delta$) graph (Fig. 2b), where all curves reached linear relations at ~70 nm.

### Young's modulus and fracture strength
The indentation-obtained Young's moduli of the 1 L BN with different isotope contents are compared in the histograms and corresponding normal distributions in Fig. 2c. The mean $E^{2D}$ values with standard errors were 299 ± 5 ($N = 24$), 284 ± 3 ($N = 45$), and 281 ± 3 ($N = 23$) N m$^{-1}$

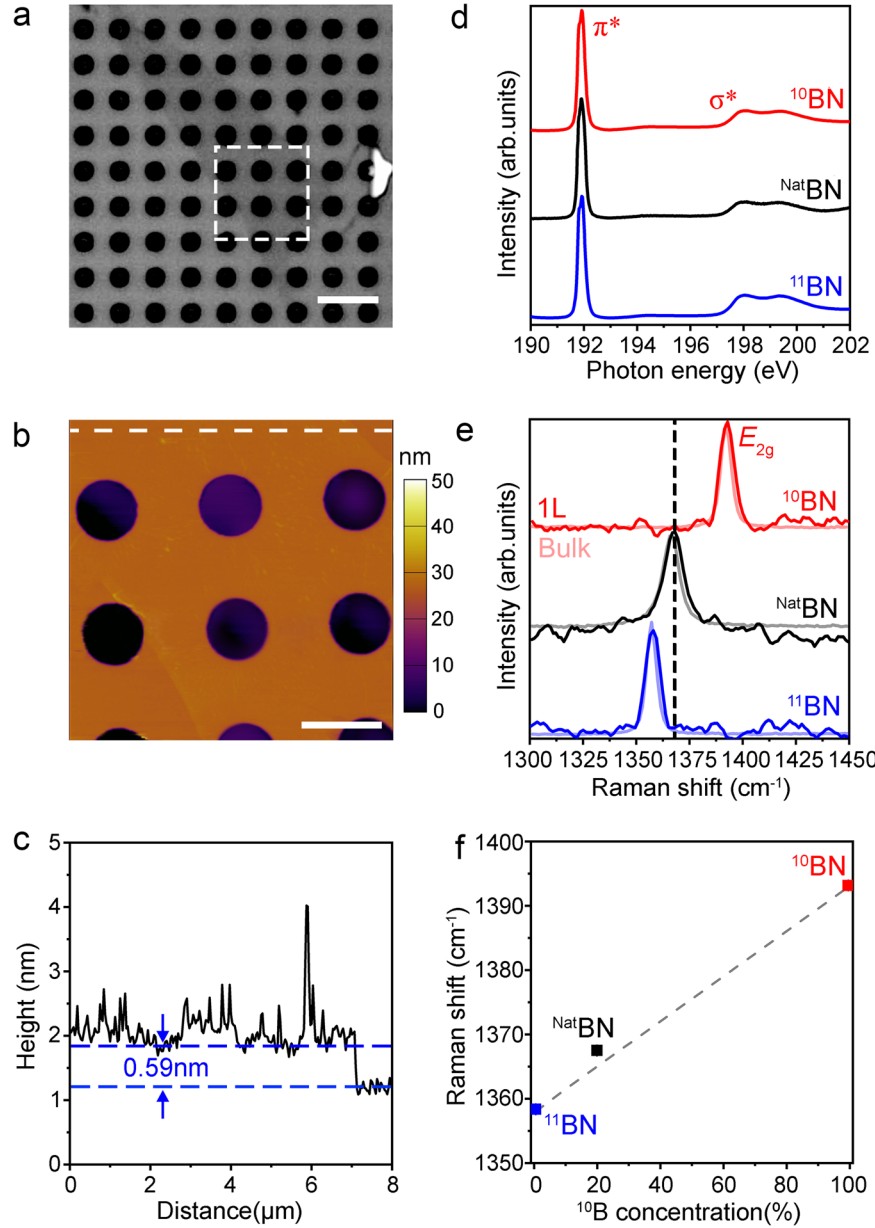

**Fig. 1 | Materials and characterizations. a** Optical microscopy image of a 1 L [11]BN on a SiO₂/Si substrate with micro-wells of 1.6 μm in diameter. **b** repulsive tapping mode AFM image of the [11]BN sheet marked in the square of (**a**). **c** The corresponding height trace of the dashed line in (**b**). **d** NEXAFS spectra at the B K-edge region of the isotopic BN materials. **e** comparison of the Raman spectra of suspended [10]BN, [Nat]BN, and [11]BN monolayers (darker color) and their bulks crystals (lighter color and at -1/50 intensity). **f** Raman frequency dependence on [10]B concentration. Scale bars 5 μm in (**a**) and 2 μm in (**b**).

for 1 L [10]BN, [Nat]BN, and [11]BN, respectively. The $E^{2D}$ values with standard deviations were 299 ± 22, 284 ± 21, and 281 ± 10 N/m for 1 L [10]BN, [Nat]BN, and [11]BN, respectively. The errors were taken as a combination of random and systematic errors. The systematic errors included the calibration parameter errors and model parameter errors in the AFM nanoindentation, and the former had a relatively larger impact on the results (Supplementary Note 2 and Table 1)[27]. The corresponding volumetric Young's moduli ($E$), i.e. 894 ± 15 GPa, 851 ± 10 GPa, and 842 ± 9 GPa, respectively were calculated by dividing $E^{2D}$ by the theoretical thickness of 1 L BN (i.e. 0.334 nm). The values of 1 L [Nat]BN were consistent with those in our previous report[4] and many theoretical predictions[22,28–31]. The isotope effect is shown more clearly in Fig. 2e, where the $E^{2D}$ of 1 L [10]BN, [Nat]BN, and [11]BN closely followed a linear trend (dashed line). The significance of the differences in $E$ was confirmed by two-sample t-Test statistics (Supplementary Note 2 and Table 2). It strongly suggests that the measured difference in Young's modulus

between 1 L [10]BN and [11]BN was indeed from the isotope effect instead of statistical uncertainties.

The fracture strength values were calculated based on the fracture loads and load-displacement curves using finite element method (FEM), and the results are shown in Fig. 2d. The average strength values of the 1 L [10]BN, [Nat]BN, and [11]BN with standard errors were 75.7 ± 1.6 GPa (25.3 ± 0.5 Nm⁻¹), 73.7 ± 1.2 GPa (24.6 ± 0.4 Nm⁻¹) and 73.1 ± 1.2 GPa (24.4 ± 0.4 Nm⁻¹), respectively. The strength values of the 1 L [10]BN, [Nat]BN, and [11]BN with standard deviations were 75.7 ± 7.3 GPa, 73.7 ± 7.4 GPa and 73.1 ± 5.2 GPa, respectively. From Fig. 2f, it can be seen more clearly that the strength increases with decreased atomic mass, similar to the trend observed in Young's modulus (Fig. 2e). However, the two-sample t-Test statistics showed that the difference in strength was not significant, i.e. the possibility of chance could not be rejected (Supplementary Note 2 and Table 2). Note the measured strength of 2D materials is highly sensitive to defects, and even one

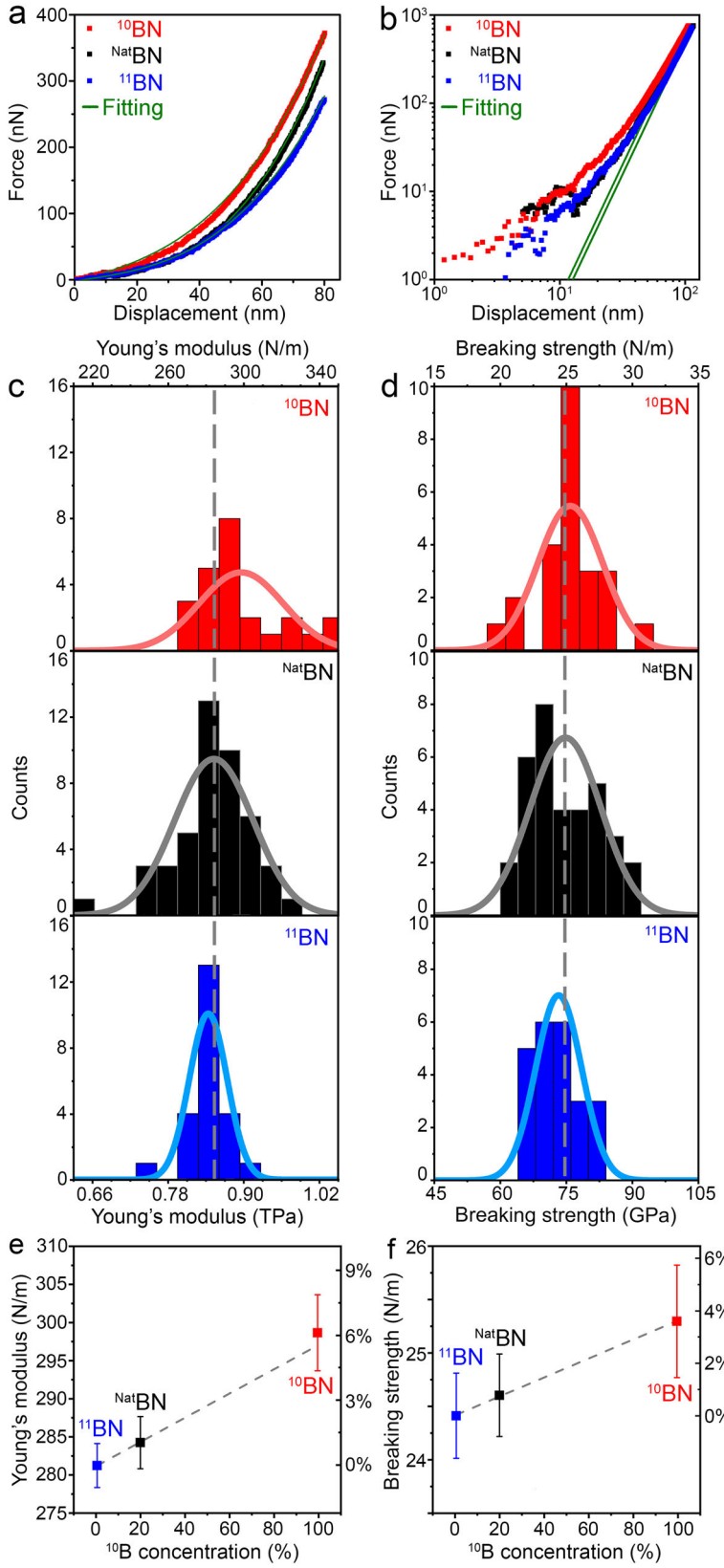

**Fig. 2 | Mechanical properties of the isotopic BN monolayers. a** Load–displacement curves and the corresponding fittings. **b** Logarithmic curves and the corresponding fittings to show the cubic behaviors at -70 nm displacement. **c** Histograms of Young's modulus and (**d**) breaking strength, where the solid lines are the fitted distribution curves for each of the sample, and the dashed lines are the mean values of $^{Nat}$BN. Young's modulus (**e**) and breaking strength (**f**) dependence on $^{10}$B contents with standard errors.

defect in the effective indentation area could cause pre-mature fracture of atomically thin materials and dramatically decrease their measured fracture strength values[32–37].

## DFT calculations

To provide theoretical understanding, we undertook ab initio density functional theory simulations (DFT) corrected with vdW interactions. The calculations unveiled that the isotope effect was caused by isotope-modified interatomic interactions and bond length variations (Fig. 3). At 0 K, the lattice constants (*a*) of 1 L $^{10}$BN and $^{11}$BN were 2.51111 Å and 2.51278 Å, respectively (Fig. 3a, b and Supplementary Table 3), a bond length difference of 0.0017 Å or -0.07%. Bader charge analysis showed that $^{10}$B had 0.0008 more electron transfer to N than $^{11}$B, justifying the shorter bond length in $^{10}$BN. The shorter lattice constant means a smaller bond length and unit cell area of $^{10}$BN (Fig. 3c and Supplementary Table 4). The DFT-deduced Young's moduli of 1 L $^{10}$BN and $^{11}$BN at 0 K were 838.6 GPa (280.1 N m$^{-1}$) and 835.3 GPa (279.0 N m$^{-1}$), respectively (Fig. 3d and Supplementary Table 3). The intrinsic fracture strength of 1 L BN is governed by its bond energy. According to DFT, the bond dissociation energy (BDE) of $^{10}$BN and $^{11}$BN at 0 K were −11.69801 and −11.70334 eV, respectively. That is, the individual $^{10}$B − N bond was weaker (Fig. 3e and Supplementary Table 4). However, the fracture strength from the AFM indentation correlates to the BDE per unit area, as the load and stress were distributed over the tip contact area under the same indentation conditions. As $^{10}$BN had a smaller unit cell area than $^{11}$BN, its BDE per area became larger than $^{11}$BN: −2.14212 *vs.* − 2.14027 eV Å$^{-2}$ (Fig. 3f). This analysis suggests that 1 L $^{10}$BN should have a larger fracture strength than 1 L $^{11}$BN. These DFT calculated Young's modulus and strength (i.e. BDE per unit area) at 0 K were in line with our experimental trend.

The isotope effects on interatomic interaction and bond length could change at elevated temperatures, especially for light elements, so we also conducted DFT simulations at 300 K. The bond length of 1 L $^{10}$BN remained shorter than that of $^{11}$BN, though the difference decreased from -0.07% at 0 K to -0.02% at 300 K (Fig. 3b). We used Synchrotron-based XRD at room temperature to experimentally measure the isotope effect on the in-plane lattice constant of bulk *h*BN: 2.5038 ± 0.0002 Å for $^{10}$BN and 2.5051 ± 0.0006 Å for $^{11}$BN (Fig. 4b and Supplementary Table 4). These XRD results agreed with the trend from our DFT calculations on 1 L BN. The theoretical Young's modulus of $^{10}$BN was higher than that of $^{11}$BN at 300 K (Fig. 3c). The BDE per unit area of $^{10}$BN was still larger than that of $^{11}$BN at 300 K, but the difference reduced to less than 0.02% (Fig. 3f and Supplementary Table 5). Note our DFT calculations did not include anharmonic vibration.

## Anomalous isotope effects

The anomalous isotope effect was caused by the changed nuclear charge distribution rather than mass difference. A $^{10}$B atom has a smaller mean square effective nuclear charge radius than a $^{11}$B atom, attributed to the specific cluster structure of the nucleus in these two

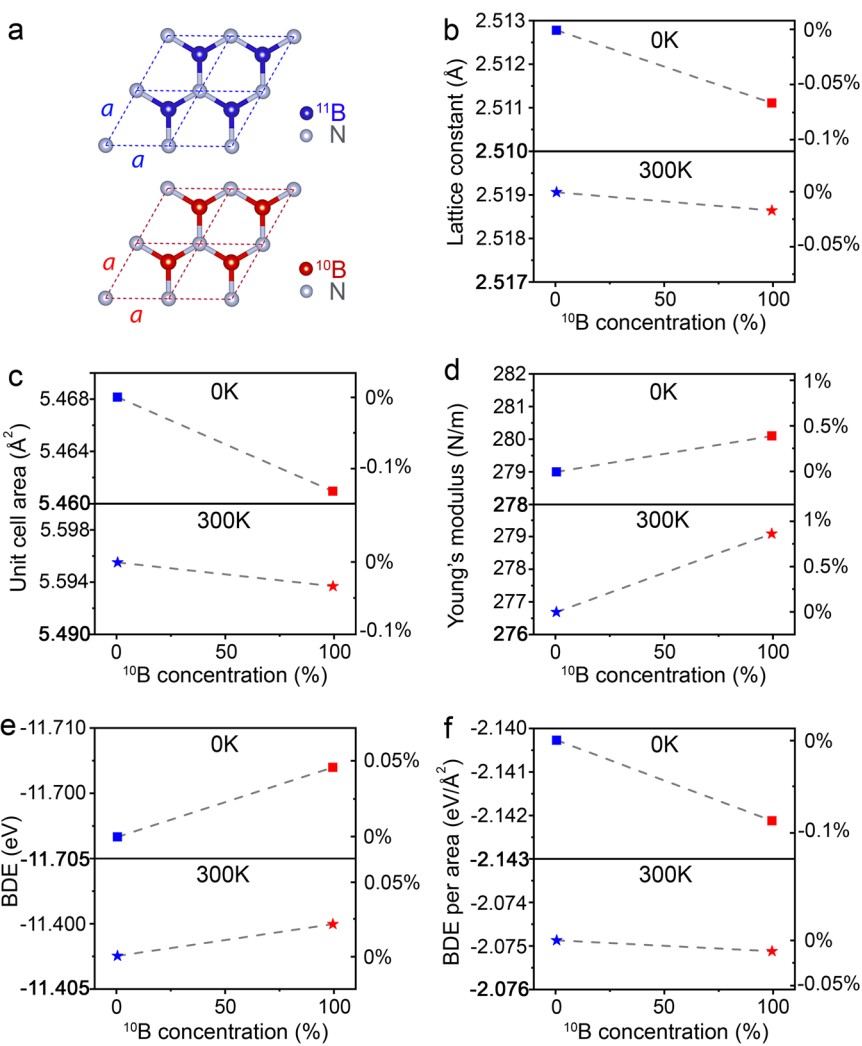

**Fig. 3 | DFT simulations on the bonding structure and mechanical properties of isotopic 1 L BN. a** The unit cells of isotopic $^{10}$BN and $^{11}$BN monolayers. **b** The lattice constant *a*, (**c**) unit cell area, (**d**) Young's modulus, (**e**) BDE, and (**f**) BDE per unit cell area at 0 K and 300 K.

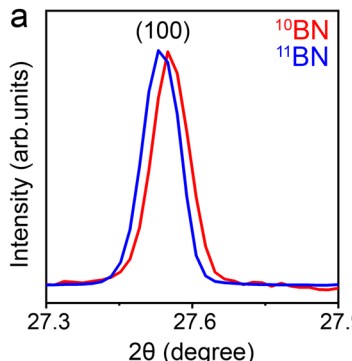
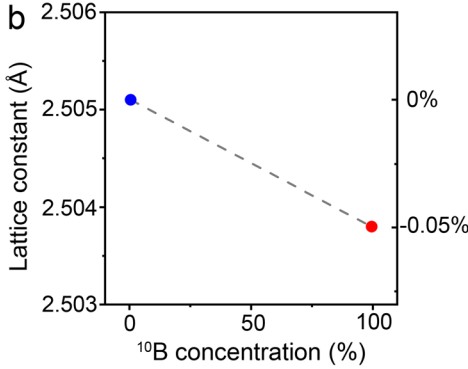

**Fig. 4 | Synchrotron-based XRD. a** The (100) XRD peaks of the isotopically pure *h*BN crystals at 300 K and (**b**) the corresponding lattice constants.

isotopes. That is, a deuteron binds with two alpha-like clusters in ${}^{10}$B less strongly than a triton in ${}^{11}$B so that the nucleus of ${}^{10}$B pulls the electrons closer, resulting in a smaller atomic radius[38] and hence a shorter bond length to N. This is supported by our DFT results. In addition, the larger effective nuclear charge of ${}^{10}$B makes the interaction potential of B − N narrower, increasing the bond stiffness[39]. This nuclear charge effect from B is opposite to those from most other isotopes of light elements, except carbon (C). More importantly, this anomalous isotope nuclear charge effect is competing with the commonly considered mass effect, i.e. a smaller nuclear mass results in weaker interaction and lower mechanical properties. As shown by our BDE results, the bond strength was dominated by the isotopic mass effect, i.e. lighter ${}^{10}$B has a higher zero-point vibrational energy level and hence ${}^{10}$B − N requires less energy to break up. However, the nuclear charge effect reduces the bond length of ${}^{10}$B − N, i.e. the BDE per unit cell for ${}^{10}$BN is larger than that of ${}^{11}$BN. Consequently, the fracture strength of the 1 L ${}^{10}$BN is larger.

Temperature makes the system more complicated because of varied vibrational states and bonding energies for different isotopic masses. A heavier isotope gains more energy than a lighter isotope during temperature increase, leading to a larger vibrational state change for the heavier isotope, whose bonding energy decreases quicker than that of the lighter isotope (Supplementary Note 3)[9,10]. In total, increased temperature can change the isotopic BN bonding and make the isotope effect on stiffness more pronounced at 300 K, as shown in our experiment and theoretical calculations. Our DFT calculations at 0 K and room temperature confirmed that the BDE of ${}^{11}$B − N dropped quicker with temperature (Fig. 3e). However, the difference in lattice constant (i.e., bond length) also decreased, leading to smaller differences in unit cell area and consequently in BDE per unit cell values (Fig. 3b, c, f). As a result, it was more difficult to measure the difference in the fracture strength than Young's modulus of isotopic BN at room temperature than cryogenic temperature.

Although only two stable isotopes of B were used in this study, our measured isotope effects on the mechanical properties of single-atom-thin BN were fully consistent with the DFT calculations. The prominent nuclear charge effect observed in isotopic B could also be present in nearby elements, such as C, N and O. In the cluster structure model of nucleus, ${}^{12}$C can be viewed as three alpha-like clusters that are tightly bound together by the strong nuclear force; ${}^{13}$C consists of three alpha-like clusters and one neutron[40]. Therefore, the mean square effective nuclear charge radius of ${}^{13}$C should be slightly larger than that of ${}^{12}$C due to the additional neutron, giving rise to shorter bonds in ${}^{12}$C than ${}^{13}$C[41]. The nuclei of ${}^{14}$N and ${}^{15}$N consist of three alpha-like clusters and a deuteron and a triton, respectively; ${}^{16}$O has four alpha-like clusters, and ${}^{17}$O and ${}^{18}$O have alpha-like clusters plus one and two additional neutrons, respectively. Therefore, these nucleus structures could lead to similar anomalous isotope effects in C, N and O that the nuclear charge

effect outweighs the mass effect, though the anomalous isotope effect may become weaker. These anomalous effects could be experimentally verified on ${}^{12}$C and ${}^{13}$C graphene readily, though it has been theoretically predicted by MD simulations that the Young's modulus[42,43] and fracture strength[43] of graphene decreased with the increase of carbon isotope mass, i.e. ${}^{12}$C, ${}^{13}$C and ${}^{14}$C. In addition, we started to synthesize *h*BN single crystals with ${}^{14}$N and ${}^{15}$N so that the mechanical measurements could be extended to one-atom-thin ${}^{10}$B${}^{14}$N, ${}^{10}$B${}^{15}$N, ${}^{11}$B${}^{14}$N and ${}^{11}$B${}^{15}$N for more generalized conclusions. On the other hand, even with the presence of nuclear charge effect, lighter and heavier elements are unlikely to show this anomalous isotope effect due to the dominating mass effect and nuclear size effect, respectively.

In summary, we were able to measure the subtle isotope effect on the mechanical properties of one-atom-thin *h*BN by AFM-based nanoindentation. Monolayer *h*${}^{10}$BN showed a higher Young's modulus and larger fracture strength than *h*${}^{11}$BN, and the two mechanical properties seemed to be linearly dependent on the isotope content. This anomalous isotope effect suggested that the isotopic nuclear charge effect was more influential than the isotopic mass effect in atomically thin *h*BN. This work demonstrated 2D materials as a group of close-to-ideal platforms to experimentally study even very subtle mechanical phenomena and mechanisms using traditional stress-strain based techniques.

## Methods

### Materials, fabrication and characterisation

The suspended atomically thin BN was prepared using mechanical exfoliation by the Scotch tape method[44,45]. The Raman spectra were obtained with a 514.5 nm laser and with an objective lens of 100× (numerical aperture 0.90) using a Renishaw inVia micro-Raman system. Prior to all Raman measurements, calibrations were performed using the Raman band of Si (520.5 cm$^{-1}$). The NEXAFS measurements were performed in the ultra-vacuum chamber (10$^{-10}$ mbar) at the magic incidence angle at the soft X-ray beamline, Australia. The XRD patterns were collected TLS 01C2 beamline of the NSRRC, Taiwan. The X-ray wavelength was 1.0332 Å (12 keV), and a MAR345 imaging plate detector was used. The powder samples were pasted between two Scotch tapes[46].

### Mechanical measurements

The nanoindentation tests were performed using a Cypher AFM. In the Z-axis, the noise floor was < 15 pm, and the accuracy of displacement was estimated to be 0.7% (Supplementary Note 1). The cantilevers with diamond tips were used as indenters as atomically thin BN has high strength and could deform regular silicon tips during indentation. Six cantilevers were used for the tests, with apex radii of 3.3, 4.9, 5.2, 5.6, 6.3, and 20.2 nm, derived from indenting high-quality boron nitride with known mechanical properties[4] and double-checked by a

transmission electron microscope. The results from different tips resulted in very similar strength and stiffness values from the tested 2D BN. The mechanical characteristics of the cantilevers were determined by a combination of Sader and thermal noise methods. The loading velocity for all indentation tests was constant ($0.5\,\mu m\cdot s^{-1}$). The indentation tests were performed only on large flakes to avoid the slippage of atomically thin BN on substrates. Also, if large hysteresis in loading-unloading curves was detected, the curves were excluded from consideration. To calculate Young's moduli, the loading/unloading indentation curves were fitted by Eq. (1) till at least 80 nm of deflection, as all curves should be in the cubic regime. All indentation tests were conducted in ambient conditions.

### Finite element analysis

The computations were performed using the commercial nonlinear finite element code ABAQUS. The indenters (monocrystal diamond tips) were modeled as rigid spheres. The nanosheets were modeled as axisymmetric circular shells with a radius of 350, 650, 700, and 800 nm. The initial thickness for isotopically pure and natural BN nanosheets were assigned $0.334 \cdot N$ nm, where N is the number of layers. Two-node linear axisymmetric shell elements (SAX1) were employed with mesh densities varying linearly from 0.1 nm (centre) to 5.0 nm (outermost). The interactions between the indenter tip and nanosheet were modeled by a frictionless contact algorithm. Displacement-controlled loading was applied to the indenter with an incremental displacement of 0.1 nm per load step. The constitutive behaviours of different BN samples were assumed to be nonlinear elastic, and thus expressed as:

$$\sigma = E\varepsilon + D\varepsilon^2 \tag{2}$$

where $D$ is the third-order elastic constant; $\varepsilon$ is an applied strain. The Young's moduli and $D$ of BN samples were obtained from the experimental results. The nonlinear elastic constitutive behaviour was implemented in ABAQUS using an equivalent elastic-plastic material model as in the previous work[4]. The load–displacement curves obtained from the simulations were compared with the corresponding experimental curves. The points at which fracture took place in the simulated curves were identified based on the fracture loads from experiments, in order to compute the corresponded fracture strength values. Subsequently, the fracture strength was obtained as a volume average of the stress values of the elements that were directly underneath the indenter at the corresponding loading steps.

### DFT calculations

All first-principles calculations were carried out based on DFT using VASP 5.4.4 code within the PAW-PBE scheme. The generalized gradient approximation along with the DFT-D3 (Grimmer) functional was used, with a well-converged plane-wave cutoff of 800 eV. The atomic coordinates were allowed to relax until the forces on the ions were less than $1 \times 10^{-3}$ eV $\mathring{A}^{-1}$ under the conjugate gradient algorithm. For sufficient accuracy, the electronic convergence was set to $1 \times 10^{-8}$ eV. The reduced Brillouin zone was sampled with a $\Gamma$-centered k-grid meshes of $24 \times 24 \times 1$ and $24 \times 24 \times 6$ for monolayer/bilayer and bulk $h$BN, respectively, which was sufficient within the k-grid mesh density lower than 0.02. A 15 $\mathring{A}$ vacuum space was used in all calculations. For different B isotopes, we used different atomic masses, i.e. 10.013 a.u. and 10.811 a.u. for $^{10}$B and $^{11}$B, respectively. The lattice parameters and atomic positions of $^{10}$BN and $^{11}$BN unit-cells were fully relaxed using the cutoff energy of 800 eV, which was sufficient for the elemental B or N with the maximum cutoffs of 318 and 400 eV. The force and electronic convergence on each atom were set to be $1 \times 10^{-3}$ eV $\mathring{A}^{-1}$ and $1 \times 10^{-8}$ eV, respectively. For the monolayer, only the lattice parameters and atomic positions in the two-dimensional plane were relaxed within the fixed symmetry of P-6M2 (No. 183). The DFT-D3 (Grimme) functional was also used, and the Gaussian smearing was utilized with the SIGMA value of 0.01. To calculate the elastic constants, we chose seven compression-tensile structures ranging from −1.5% to 1.5% with an interval of 0.5% compared with the optimized lattice. Phonon spectra were calculated using the VASP 5.4.4 and Phonopy 1.12.2 codes with a supercell of $2 \times 2 \times 1$. To calculate the sliding energy, we chose nine intermediate structures along the periodic direction. To include thermal effects, we calculated the energy of phonons using the quasi-harmonic approximation. Thermal properties such as free energy of phonons, heat capacity, and entropy were computed under temperatures ranging from 0 K to 300 K with compression-tensile structures to confirm the most stable structures with finite temperatures. We further calculated the free energy of compression-tensile structures and obtain the mechanical properties. The corresponding input and structural files are provided (see Supplementary Information).

### Data availability

The data that support the findings of this study are available from https://doi.org/10.6084/m9.figshare.22884224.

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

## Acknowledgements

Support for hBN crystal growth was provided by the Office of Naval Research Award number N00014-20-1-2474. This research was partly undertaken on the soft x-ray beamline at the Australian Synchrotron, Victoria, Australia. LHL thanks the $^{Nat}$BN crystals from Prof. Takashi Taniguchi and Prof. Kenji Watanabe from NIMS, Japan and valuable discussions with Prof. Matthew Barnett at Deakin University, Australia. EJGS acknowledges computational resources through CIRRUS Tier-2 HPC Service (ec131 Cirrus Project) at EPCC (http://www.cirrus.ac.uk) funded by the University of Edinburgh and EPSRC (EP/P020267/1); ARCHER UK National Supercomputing Service (http://www.archer.ac.uk) via Project d429. EJGS also acknowledges the Spanish Ministry of Science's grant program "Europa-Excelencia" under grant number EUR2020-112238, the EPSRC Open Fellowship (EP/T021578/1), and the Edinburgh-Rice Strategic Collaboration Awards for funding support.

## Author contributions

L.H.L. conceived the project; E.J. and J.H.E. synthesized the isotopic hBN crystals; A.F., L.H.L., and W.G. prepared the atomically thin BN; A.F. did the mechanical measurements and data analysis; R.Z. and D.Q. did the FEM simulations; H.L. and X.W. did the DFT calculations; H.S.S. measured Synchrotron XRD; L.H.L. and Q.C. did NEXAFS; L.H.L., A.F., E.J.G.S., J.H.E. discussed the results; L.H.L. and A.F. wrote the manuscript.

## Competing interests

The authors declare no competing interests.
