## [Peer Review File · Nature Communications]

Anomalous Isotope Effect on Mechanical Properties of Single Atomic Layer Boron NitrideREVIEWER COMMENTS

Reviewer #1 (Remarks to the Author):

The manuscript reports an isotope effect on the mechanical properties of monolayer hBN. This is a very interesting topic, even though it is hard to predict a similar influence in other systems.

1. AFM was employed to conduct indentation test for measuring the elasticity and strength. It will be great if a quick literature review about how to evaluate the mechanical properties of 2D materials can be added to either the main text or supplementary material.
2. What is the vertical spatial resolution of the AFM used in this manuscript? In Figure 1(c) and Figure S1(c,f), what is the uncertainty of the BN thickness measured by AFM?
3. It will be great if more details about AFM image could be provided. For example, it should be noted what kind of AFM image it is.
4. Figure 1(c) seems oversaturated, and difficult to see any details in the suspended BN. Please reset the data bar from 0-50 nm to 0-5 nm or 0-10 nm or 0-20 nm. Is it possible to show the height profile across the suspended BN? In case the suspended BN is ideally flat, the fluctuation should be smaller than 2 nm. If it is not flat enough, does the bubble geometry influence the test in terms of effective area or force?
5. In Figure 2(a), please explain more about the data fitting, especially how to tell a good fitting and the error from fitting. I am curious what the fitted curve looks like for NatBN, if the modulus has a 10% offset on purpose.
6. In the main text, the error should be clarified. Usually, the standard deviation (sd) should be following the mean value, instead of the standard error (SE). For example, according to the equation of SE, if 100,000 tests were done with 10BN, the SE in the manuscript should be less than 0.02. While the sd should be still on the order of 20, reflecting the data distribution. I strongly suggest replacing the SE with standard deviation in all figures and main text, which is consistent with other reports.
7. It will be great if some SEM/TEM images are shown in the manuscript to support the assumption of defect-free samples.

Reviewer #2 (Remarks to the Author):

The paper deals with the determination of the mechanical properties particularly of single-atom-thin isotopically pure hexagonal boron nitride (hBN). The results make sense and are consistent with the expected isotopic effect. The materials has been classically prepared via the scotch exfoliation of crystalline BN materials. However, as in previous papers describing the thickness of the layer and thus its single atom nature is deduced from AFM measurements. Therefore, since boron and nitrogen atoms are necessarily bonded to side atoms or groups, it is important to characterize the surface in terms of functional groups which could be present. Thus, I recommend a thorough characterization of the "surface" (the said single atom material) in order to confirm the interpretations before publication.

Reviewer #3 (Remarks to the Author):

The manuscript explores the subtle isotope effect on the mechanical properties of single-layered hexagonal boron nitride (h-BN) with both experimental tests and DFT simulation. While the manuscript is well-organized and presents interesting results, there are a few areas that could be improved:

1. The detailed description of the limitations of the study should be described more in the manuscript, which may affect the generalizability of their results. The author listed several reasons why hBN was chosen in this study. However, with only two stable isotopes, hBN can only provide two effective data points (^{10}BN , ^{11}BN), which seems to be insufficient to draw a conclusion. On the other hand, elements with more stable isotopes may be hard to form a 2D-structure material.

2. 2D materials typically have fewer disorders. But it doesn't mean there will not be any defect in the effective measuring area. Since the defect can greatly affect the mechanical performance, it could make the results more convincing to show atomic resolution images of the three samples.

3. The manuscript may lack a more comprehensive comparison to experimental data, which may limit the impact of this study. While the author briefly mentioned some previous experimental studies about the isotope effect, they do not provide a detailed comparison of their results to these studies. As for simulation results, it could provide important insights into the accuracy of their simulation approach if the authors can compare it with other research, such as some simulation results on graphene.

4. The authors did not fully discuss how detailed description of the computational details of their DFT simulations. This lack of information makes it difficult for readers to assess the accuracy and reliability of the DFT results.

5. The AFM results indicate the thickness of 1L BN is 0.59 nm, while the authors used the theoretical value (0.334 nm) in their calculation. The author may need to explain why the measured result is much higher than the theoretical value. If the measurement results are within the error bar, then how can the author make sure the sample is mono-layer instead of double-layer?

Overall, while the manuscript presents interesting results and is well-written, there are a few major revisions that could be improved to enhance the impact and generalizability of the study.

We are grateful for the valuable comments and feedback from the Reviewers to improve the quality and clarity of our manuscript. Below is our detailed point-by-point response:

REVIEWER COMMENTS

Reviewer #1 (Remarks to the Author):

The manuscript reports an isotope effect on the mechanical properties of monolayer hNB. This is a very interesting topic, even though it is hard to predict a similar influence in other systems.

Change: We thank the Reviewer for the useful comments and suggestions. To extend the finding to “a similar influence in other systems”, the following paragraph has been added to the main text:

P.16: Although only two stable isotopes of B were used in this study, our measured isotope effects on the mechanical properties of single-atom-thin BN were fully consistent with the DFT calculations. The prominent nuclear charge effect observed in isotopic B could also be present in nearby elements, such as C, N and O. In the cluster structure model of nucleus, ^{12}C can be viewed as three alpha-like clusters that are tightly bound together by the strong nuclear force; ^{13}C consists of three alpha-like clusters and one neutron⁴⁰. Therefore, the mean square effective nuclear charge radius of ^{13}C should be slightly larger than that of ^{12}C due to the additional neutron, giving rise to shorter bonds in ^{12}C than ^{13}C ⁴¹. The nuclei of ^{14}N and ^{15}N consist of three alpha-like clusters and a deuteron and a triton, respectively; ^{16}O has four alpha-like clusters, and ^{17}O and ^{18}O have alpha-like clusters plus one and two additional neutrons, respectively. Therefore, these nucleus structures could lead to similar anomalous isotope effects in C, N and O that the nuclear charge effect outweighs the mass effect, though the anomalous isotope effect may become weaker. These anomalous effects could be readily verified on ^{12}C and ^{13}C graphene experimentally, though it has been theoretically predicted by MD simulations that the Young's modulus^{42,43} and fracture strength⁴³ of graphene decreased with the increase of carbon isotope mass, *i.e.* ^{12}C , ^{13}C and ^{14}C . In addition, we started to synthesize hBN single crystal with ^{14}N and ^{15}N so that the mechanical measurements could be extended to one-atom-thin $^{10}\text{B}^{14}\text{N}$, $^{10}\text{B}^{15}\text{N}$, $^{11}\text{B}^{14}\text{N}$ and $^{11}\text{B}^{15}\text{N}$ for more generalized conclusions. On the other hand, even with the presence of nuclear charge effect, lighter and

heavier elements are unlikely to show this anomalous isotope effect due to the dominating mass effect and nuclear size effect, respectively.

1. AFM was employed to conduct indentation test for measuring the elasticity and strength. It will be great if a quick literature review about how to evaluate the mechanical properties of 2D materials can be added to either the main text or supplementary material.

Changes: A brief literature review / introduction on how to evaluate the mechanical properties of 2D materials has been added to the Supplementary Material, Note S1.

Main text: P.7: *More details of the method are provided in the Supplementary Note S1.*

Supplementary Note S1: Measurements of mechanical properties of 2D nanomaterials using AFM

Atomic force microscope (AFM) has been the mostly used method to measure the mechanical properties of 2D materials^{1,4-6} due to the ease of accessibility as well as high reliability, though other methods such as in-plane stretching^{7,8} and bulge method^{9,10} have also been demonstrated. The AFM method was first used on graphene by C. Lee et al. in 2008¹. In the method, the 2D materials are suspended over microwells or trenches pre-fabricated on substrates. Importantly, the 2D sheets must be securely attached to the edge of the microwell or trench to prevent sliding during indentation. The AFM cantilever with a sharp tip at its end applies loads at the centre of suspended regions of the 2D materials. The material of the indenter or AFM tip is ideally stiffer than the 2D material (e.g. diamond tip for graphene and BN measurement). Also, the tip radius should be much smaller than the radius of the suspended 2D materials. The load-displacement relations provide information on the elastic properties and fracture force of the 2D materials.

To determine the Young's modulus, Eq.S1 is used, which is based on a combination of the solutions from 1) the linear behavior of 2D materials under small stress or indentation depths derived by Wan et al.¹¹, and 2) for large stresses, where load varies as the cube of displacement derived by Komaragiri et al.¹²:

$$F(\delta) = \pi\sigma_0^{2D}\delta + \left(\frac{q^3E^{2D}}{a^2}\right) * \delta^3, \quad (S1)$$

This equation is applied to fit the load-displacement $F(\delta)$ curves from AFM indentation. The first linear term of the load-displacement curve depends only on 2D pre-tension (σ_0^{2D}) due to axial tensions in the 2D materials, given fixed radii of the tip and the suspended 2D materials (a). In this linear region, the initial load and small stretch flatten the suspended 2D materials by removing wrinkles and unevenness. The pre-tension value mostly depends on the sample itself and its preparation process. This term could also include bending modulus, but this factor is negligible at the atomic thickness. With further increase in the load, the second term of Eq. S1 starts to play an important role. The load–displacement relation becomes cubic dependent and reflects the stiffness of the suspended 2D materials so that the 2D Young’s modulus (E^{2D}) value can be obtained. This term also depends on Poisson’s ratio (ν) of the 2D material which is included in eq. S1 as a dimensionless constant $q = 1/(1.05 - 0.15\nu - 0.16\nu^2)$. In this analysis, only pre-tension and Young’s modulus are variables in fitting the load-displacement curves. Later, this equation was further developed by Lin et al.(2013)¹³. The updated relation considers starting (zero) coordinates (f_0 and δ_0) as additional variables that could improve the accuracy of the calculated Young’s modulus by removing the manual determination of the zero point (Eq.1 in main text). The accuracy of the elastic properties from the load-displacement data fitting relies on whether the curve has reached the transformed Schwerin (cubic) term¹⁴ in Eq.1 (main text)¹⁵. High-quality graphene and 2D BN can typically withstand significant levels of strain to reach this region without failure.

To obtain the fracture strength of 2D materials, the measured elastic modulus, fracture load, and load-displacement curves are normally analysed by finite element method (FEM), as the strength value calculated analytically is overestimated due to the lack of nonlinear component, which is especially critical under high loads prior fracture.

The above AFM indentation method and analysis of the load-displacement curves have been successfully applied for graphene and other 2D materials, such as BN and transition metal dichalcogenides (TMDCs)^{1,6}.

2. What is the vertical spatial resolution of the AFM used in this manuscript? In Figure 1(c) and Figure S1(c,f), what is the uncertainty of the BN thickness measured by AFM?

Response: In this work, we used a Cypher AFM which had a high vertical special resolution, *i.e.*, the noise floor in Z-axis <15 pm. We estimated the uncertainty in the AFM piezoelectric stage displacement in the vertical direction to be 0.7% using standard samples of known height. The results are already provided in the **Supplementary Materials note S2**.

On the other hand, the thickness of the BN (*e.g.*, 1 Layer vs. 2L) could also be determined based on the load-displacement curves from the AFM indentation, as the 2D Young's modulus and fracture load of 2L BN double those of 1L.

Changes: We have added the noise floor in the methodology section.

P.17: "In the Z-axis, the noise floor was <15 pm, and the accuracy of displacement was estimated to be 0.7% (Supplementary note S1)."

3. It will be great if more details about AFM image could be provided. For example, it should be noted what kind of AFM image it is.

Changes: The details on the AFM images have been added to the captions of Figures 1 and S1.

Main text: Figure 1 | Materials and characterizations. a, Optical microscopy image of a 1L 11BN on a SiO₂/Si substrate with micro-wells of 1.6 μm in diameter. b, repulsive tapping mode AFM image of the 11BN sheet marked in the square of (a)..."

P.5: The corresponding atomic force microscopy (AFM) image obtained in repulsive tapping mode shows a thickness of 0.59 nm (Fig. 1b and c).

In Supplementary materials:

Figure S1| Characterization of BN nanosheets. Optical microscopy images of a, a 1L ^{Nat}BN and d, a 1L ¹⁰BN on a SiO₂/Si substrate with micro-wells of 1.6 μm in diameter. Repulsive tapping mode AFM images of b, the ^{Nat}BN nanosheets, and e, the ¹⁰BN nanosheets marked in

the square of (a) and (d), respectively. c, and f, the corresponding height traces of the dashed lines for ^{Nat}BN and ^{10}BN samples, respectively, confirming the monolayer thickness.

4. Figure 1(c) seems oversaturated, and difficult to see any details in the suspended BN. Please reset the data bar from 0-50 nm to 0-5 nm or 0-10 nm or 0-20 nm. Is it possible to show the height profile across the suspended BN? In case the suspended BN is ideally flat, the fluctuation should be smaller than 2 nm. If it is not flat enough, does the bubble geometry influence the test in terms of effective area or force?

Response: The height ranges of the AFM images in Figure 1(c) and Figure S1 (e) were chosen so that both the suspended and non-suspended regions of the monolayer BN could be seen. An example of 0-20 nm range AFM image is shown below. Please note that the suspended parts of 2D materials are normally below the surface of the micro-wells due to the attachments to the inner walls of the microwells. According to the height profile below, this monolayer BN was 20 nm beneath the surface of the micro-well. The surface fluctuation was about 1 nm in the suspended region. As described in the literature review / introduction above and Supplementary Note S1, the surface fluctuation of the suspended 2D materials is already considered as the zero-point and pre-tension (*i.e.*, $\pi\sigma_0^{2D}\delta$ in Eq. S1) in the calculation of the mechanical properties, so it won't affect the mechanical results.

Changes:

P.5: Examples of 1L ^{10}BN and $^{\text{Nat}}\text{BN}$ as well as the cross-section of the suspended region of the 1L ^{11}BN can be found in Supplementary Materials Figure S1-2.

The profile of the suspended region of the ^{11}BN monolayer has been added to Supplementary Materials:

Figure S2| Characterization of a suspended region of a monolayer BN. a, Repulsive tapping mode AFM image of the ^{11}BN nanosheet. b, the corresponding height trace of the dashed line in (a) representing the suspended part of the monolayer BN.

5. In Figure 2(a), please explain more about the data fitting, especially how to tell a good fitting and the error from fitting. I am curious what the fitted curve looks like for NatBN, if the modulus has a 10% offset on purpose.

Response: The explanation on the AFM indentation and data fitting has been described in the literature review / introduction above and added to Supplementary Note S1. The graph below

shows how $\pm 10\%$ in elastic modulus hypothetically affect the fitting. An important fitting factor is the determination of the zero point or the point where an indenter starts stretching the tested membrane, and the resultant error from this factor in the elastic modulus could be as much as 10%. This error was minimized by the additional fitting parameters in the equation developed by Lin et al. (2013) (Ref. 13) instead of manual determination of the zero point in other publications.

6. In the main text, the error should be clarified. Usually, the standard deviation (sd) should be following the mean value, instead of the standard error (SE. For example, according to the equation of SE, if 100,000 tests were done with ^{10}BN , the SE in the manuscript should be less than 0.02. While the sd should be still on the order of 20, reflecting the data distribution. I strongly suggest replacing the SE with standard deviation in all figures and main text, which is consistent with other reports.

Response: The standard deviations for ^{10}BN and ^{11}BN monolayers are already provided in the Supplementary Materials Table S1. In the revision, we added the values for $^{\text{Nat}}\text{BN}$ in the table. Furthermore, the histogram distributions of all data are provided in Fig.2c and d in the Main text.

Change: The standard deviation for $^{\text{Nat}}\text{BN}$ added to Supplementary Table S1:

Sample	x	Parameter	N	Mean, [N/m]	s_x , [N/m]	Variance, [N ² /m ²]	SE,[N/m]
1L ¹⁰ BN	1	Young's modulus	24	298.66	22.20	493.06	4.53
		Fracture strength	24	25.30	2.45	6.00	0.50
1L ^{Nat} BN	2	Young's modulus	45	284.25	20.83	433.92	3.11
		Fracture strength	45	24.60	2.47	6.09	0.37
1L ¹¹ BN	3	Young's modulus	23	281.23	10.04	100.75	2.09
		Fracture strength	23	24.41	1.74	3.04	0.36

where, s_x – standard deviation, SE –standard error.

7. It will be great if some SEM/TEM images are shown in the manuscript to support the assumption of defect-free samples.

Response:

(1) The samples were not defect-free; instead, it was stated in the manuscript (P.2) “... as well as free of in-plane defects in the effective measuring area”. Due to the small size of the cantilever tip, the effective measuring area in the indentation is normally less than 20 nm in radius [*Science* 321, 385 (2008)]. The effective indentation area is slightly larger than the contact area between AFM tip and suspended 2D materials and where most of the strain is concentrated.

(2) SEM is not effective in seeing defects in the monolayer BN.

(3) Comprehensive TEM (including SAED) and EBSD analyses of the bulk h¹⁰BN and h¹¹BN crystals that used for the mechanical exfoliation of the atomically thin isotopic BN have been

conducted previously, *i.e.*, *Communications Physics* 2, 43 (2019) or Ref.18. In this paper, it stated “Using TEM, the single crystal domain was found to have areas a few tens of microns across which are free of defects.” This Ref is already cited in the main text: “The ^{10}BN and ^{11}BN bulk crystals were grown by the nickel-chromium solvent method at atmospheric pressure,¹⁶ and the single crystal domains without defects were a few tens of microns¹⁸”. So compared to the <20 nm effective measuring radius, the chance to encounter defects during indentation was very slim.

(4) The TEM analysis of monolayer ^{10}BN and ^{11}BN was reported in our previous publications, *e.g.*, *Physical Review Letters* 125, 085902 (2020) in which the SAED and dark-field TEM image showed a single crystalline structure over $2\ \mu\text{m}$ region of the mechanically exfoliated atomically thin ^{10}BN and ^{11}BN , consistent with the result from *Communications Physics* 2, 43 (2019) that “the single crystal domain was found to have areas a few tens of microns across which are free of defects”.

(5) Most importantly, the AFM indentation-based mechanical measurement is the best way to show the defect levels in 2D materials. This is because even one defect in the effective measuring area could cause pre-mature fracture of a one-atom thin material and dramatically decrease the measured fracture strength. This effect has been mentioned in many papers, *e.g.*, *Nature Communications* 5, 3186 (2014) (or see Ref. 32-37). That is, the close-to theoretical fracture strength values measured from the monolayer ^{10}BN and ^{11}BN demonstrate that there was no decisive effect of defect on our mechanical measurements.

(6) The bulk $^{\text{Nat}}\text{BN}$ and isotopic BN crystals used for exfoliating atomically thin samples were grown by different methods in different labs, *i.e.*, Ba–BN solvent method under high-pressure in Japan and nickel-chromium solvent method at atmospheric pressure in US, respectively. The $^{\text{Nat}}\text{BN}$ from Japan has been used in numerous studies and contains extremely low density of defects, and the mechanical properties of its monolayer have been systematically studied by us [*Nature Communications* 8, 15815 (2017)]. The monolayer ^{10}BN and ^{11}BN showed differences in the average fracture strength are less than 3% compared to that of the $^{\text{Nat}}\text{BN}$. The small difference measured from one-atom-thin material is unlikely to be caused by defects, as a mono-vacancy in the effective measuring area could reduce the fracture strength by 20% or more.

Changes: We have made the following changes in the Main text

P.2: This can be attributed to less disorder due to small numbers of atoms, no surface defects due to their 2D nature, simple chemical composition, as well as free of in-plane defects in the effective measuring area that is normally less than 20 nm in radius in atomic force microscope (AFM) based indentation method.

P.5: The ^{10}BN and ^{11}BN bulk crystals were grown by the nickel-chromium solvent method at atmospheric pressure,¹⁶ and the single crystal domains without defects were a few tens of microns¹⁸.

P.5: The transmission electron microscopy (TEM) study of the suspended atomically thin isotopic BN was reported in our previous publication¹⁷.

P.7: Because the single crystal domains without defect were a few tens of microns¹⁸ and our effective measuring radius was less than 20 nm, the chance to encounter defects during the indentation was small.

P.11: Note the measured strength of 2D materials is extremely sensitive to defects, and even one defect in the effective indentation area could cause the pre-mature fracture of atomically thin materials and dramatically decrease their measured fracture strength values³²⁻³⁷.

Reviewer #2 (Remarks to the Author):

The paper deals with the determination of the mechanical properties particularly of single-atom-thin isotopically pure hexagonal boron nitride (hBN). The results make sense and are consistent with the expected isotopic effect. The materials has been classically prepared via the scotch exfoliation of cristalline BN materials. However, as in previous papers describing the thickness of the layer and thus its single atom nature is deduced from AFM measurements. Therefore, since boron and nitrogen atoms are necessarily bonded to side atoms or groups, it is important to characterize the surface in terms of functional groups which could be present. Thus, I recommend a thorough characterization of the "surface" (the said single atom material) in order to confirm the interpretations before publication.

Response: We thank the reviewer for the comment. In ideal monolayer BN, *i.e.*, no grain boundary and defect, all B-N bonds except the ones on edge are fully saturated. That is, no covalent surface functionalization could be present on our monolayer BN or affect the measured mechanical properties, because our BN monolayers were single crystalline without defect in the effective indentation area, as discussed above. For example, even if there was sp^3 C bonds / defects in the effective indentation area of graphene, its breaking strength would reduce ~14% [*Nature Communications* 5, 3186 (2014)]. The functional groups on the edge of the BN monolayers had no impact on the mechanical measurements, as the indentation was conducted at the center of the suspended regions. On the other hand, as shown in our previous publication *Adv. Funct. Mater.* 26 (45), 8356 (2016), BN could physisorb airborne molecules. However, the energy of the physisorption is dramatically lower than the energy of the covalent bonds of B-N. In other words, the physisorbed molecules even present have negligible effect on the measured mechanical properties. This conclusion is justified by many previous measurements of the mechanical properties of 2D materials, almost all of which were conducted in air.

Nevertheless, we still tried to do surface characterization as requested by the Reviewer. As the lateral size of the exfoliated atomically thin BN is small, *i.e.*, ~10 μ m, micro-FTIR in ATR mode is used to measure ~50 nm thick mechanically exfoliated BN. The FTIR spectra were the average of 128 scans using a Bruker Lumos FTIR microscope with a resolution of 4 cm^{-1} . The result below shows absorption bands of BN at 830 and 1360 cm^{-1} and Si-O-Si and Si-H bands between 950 and 1100 cm^{-1} . No C-O or C-H band is visible, suggesting relative surface cleanness of the mechanical exfoliated BN samples.

Regarding the AFM measurement to determine monolayer BN, (1) We ensured that the tapping mode was in the repulsive region; (2) we have confirmation of thickness using Raman; (3) the thickness of the BN (*e.g.*, 1L vs. 2L) was also determined based on the load-displacement curves from the AFM indentation, as the 2D Young's modulus and fracture load of 2L BN double those of 1L; (4) our group has published more than a dozen of papers on atomically thin BN such as such as *Nature Communications* 8, 15815 (2017); *Nature Communications* 9, 1 (2018); *Science Advances* 5, eaav0129 (2019); *Physical Review Letters* 125, 085902 (2020); *ACS Nano* 15, 2600 (2021); *ACS Nano* 8, 1457 (2014); *Nano Letters* 21, 3379 (2021); *Nano Letters* 15, 218 (2015); *Advanced Functional Materials* 26, 8356 (2016), and the layer number determination is a routine practice.

Reviewer #3 (Remarks to the Author):

The manuscript explores the subtle isotope effect on the mechanical properties of single-layered hexagonal boron nitride (h-BN) with both experimental tests and DFT simulation. While the manuscript is well-organized and presents interesting results, there are a few areas that could be improved:

1. The detailed description of the limitations of the study should be described more in the manuscript, which may affect the generalizability of their results. The author listed several reasons why hBN was chosen in this study. However, with only two stable isotopes, hBN can only provide two effective data points (^{10}BN , ^{11}BN), which seems to be insufficient to draw a conclusion. On the other hand, elements with more stable isotopes may be hard to form a 2D-structure material.

Response and change: We thank the Reviewer for the valuable comments.

So far, all publications on the isotope effect on various physical properties were based on **only one element with two to three isotopes**: (i) lattice parameters [*Nature* 134, 900-901 (1934); *Archives des Sciences Physiques et Naturelles* (1937); *Nature Materials* 17, 152-158 (2018); *Journal of Chemical Physics* 22, 1062-1063 (2004); *Journal of the American Chemical Society* 69, 1719-1723 (1947); *Contribution à l'étude des hydrures métalliques* 12,

305-362 (1955)], **(ii)** interlayer spacing in van der Waals bonding in crystals [*Nature Materials* 17, 152-158 (2018); *Soviet Physics Uspekhi* 5, 951 (1963); *Physical Review Letters* 124, 167402 (2020)], **(iii)** thermal conductivity [*Physical Review Letters* 124, 167402 (2020); *Communications Physics* 2, 43 (2019); *Physical Review Letters* 125, 085902 (2020)], and **(iv)** thermal expansion coefficient [*European Physical Journal B* 89, 56 (2016)].

In terms of **all** previous publications involving experiments on isotope effects on mechanical properties, only one element with **two isotopes** were involved: the ultrasound waves method [*Journal of Applied Physics* 76, 7726-7730 (1994)], the Brillouin scattering of light measurements on ^{12}C and for ^{13}C diamond crystals [*Acta Crystallographica Section B: Structural Science* 52, 232-238 (1996); *Physical Review Letters* 71, 189 (1993); *Physical Review B* 54, 3989 (1996)], the ultrasound method to measure elastic constants of ^7LiH and ^7LiD isotope crystals [*Zeitschrift für Kristallographie-Crystalline Materials* 130, 340-368 (1969); *Journal of Physics C: Solid State Physics* 15, 6321 (1982)] or indirect estimation of their elasticity through Raman spectra [*Physical Review B* 51, 8874-8877 (1995)], and even most theoretical studies on mechanical properties used only two isotopes [*Computational Materials Science* 71, 197-200 (2013); *Computational Condensed Matter* 11, 11-19 (2017); *Physical Review B* 80, 113405 (2009)].

Hence, these previous investigations support our approach to consider two isotopes in our study.

We agree that the “limitations” and “generalizability” should be addressed. We have added a new paragraph at the end of the main text:

P.16: Although only two stable isotopes of B were used in this study, our measured isotope effects on the mechanical properties of single-atom-thin BN were fully consistent with the DFT calculations. The prominent nuclear charge effect observed in isotopic B could also be present in nearby elements, such as C, N and O. In the cluster structure model of nucleus, ^{12}C can be viewed as three alpha-like clusters that are tightly bound together by the strong nuclear force; ^{13}C consists of three alpha-like clusters and one neutron⁴⁰. Therefore, the mean square effective nuclear charge radius of ^{13}C should be slightly larger than that of ^{12}C due to the additional neutron, giving rise to shorter bonds in ^{12}C than ^{13}C ⁴¹. The nuclei of ^{14}N and ^{15}N consist of three alpha-like clusters and a deuteron and a triton, respectively; ^{16}O has four

alpha-like clusters, and ^{17}O and ^{18}O have alpha-like clusters plus one and two additional neutrons, respectively. Therefore, these nucleus structures could lead to similar anomalous isotope effects in C, N and O that the nuclear charge effect outweighs the mass effect, though the anomalous isotope effect may become weaker. These anomalous effects could be readily verified on ^{12}C and ^{13}C graphene experimentally, though it has been theoretically predicted by MD simulations that the Young's modulus^{42,43} and fracture strength⁴³ of graphene decreased with the increase of carbon isotope mass, *i.e.* ^{12}C , ^{13}C and ^{14}C . In addition, we started to synthesize hBN single crystal with ^{14}N and ^{15}N so that the mechanical measurements could be extended to one-atom-thin $^{10}\text{B}^{14}\text{N}$, $^{10}\text{B}^{15}\text{N}$, $^{11}\text{B}^{14}\text{N}$ and $^{11}\text{B}^{15}\text{N}$ for more generalized conclusions. On the other hand, even with the presence of nuclear charge effect, lighter and heavier elements are unlikely to show this anomalous isotope effect due to the dominating mass effect and nuclear size effect, respectively.

P.4: ... these results contradictory to the commonly considered isotopic mass effect are attributed to the ultrafine isotopic nuclear charge effect that could be present in other elements close to B in the periodic table.

2. 2D materials typically have fewer disorders. But it doesn't mean there will not be any defect in the effective measuring area. Since the defect can greatly affect the mechanical performance, it could make the results more convincing to show atomic resolution images of the three samples.

Response and changes: The Reviewer is absolutely correct that 2D materials do have defects and defects could greatly affect their mechanical properties if they are located in the effective indentation area. Because of the high similarity of this comment to #7 comment from Reviewer 1, we have the same response and changes:

(1) The samples were not defect-free; instead, it was stated in the manuscript (P.2) "... as well as free of in-plane defects in the effective measuring area". Due to the small size of the cantilever tip, the effective measuring area in the indentation is normally less than 20 nm in radius [*Science* 321, 385 (2008)]. The effective indentation area is slightly larger than the contact area between AFM tip and suspended 2D materials and where most of the strain is concentrated.

(3) Comprehensive TEM (including SAED) and EBSD analyses of the bulk $h^{10}\text{BN}$ and $h^{11}\text{BN}$ crystals that used for the mechanical exfoliation of the atomically thin isotopic BN have been conducted previously, i.e. *Communications Physics* 2, 43 (2019) or Ref.18. In this paper, it stated “Using TEM, the single crystal domain was found to have areas a few tens of microns across which are free of defects.” This Ref is already cited in the main text: “The ^{10}BN and ^{11}BN bulk crystals were grown by the nickel-chromium solvent method at atmospheric pressure,¹⁶ and the single crystal domains without defects were a few tens of microns¹⁸”. So compared to the <20 nm effective measuring radius, the chance to encounter defects during indentation is very slim.

(4) The TEM analysis of monolayer ^{10}BN and ^{11}BN was reported in our previous publications, e.g., *Physical Review Letters* 125, 085902 (2020) in which the SAED and dark-field TEM image showed a single crystalline structure over $2\ \mu\text{m}$ region of the mechanically exfoliated atomically thin ^{10}BN and ^{11}BN , consistent with the result from *Communications Physics* 2, 43 (2019) that “the single crystal domain was found to have areas a few tens of microns across which are free of defects”.

(5) Most importantly, the AFM indentation-based mechanical measurement is the best way to show the defect levels in 2D materials. This is because even one defect in the effective measuring area could cause pre-mature fracture of a one-atom thin material and dramatically decrease the measured fracture strength. This effect has been mentioned in many papers, e.g., *Nature Communications* 5, 3186 (2014) (or see Ref. 32-37). That is, the close-to theoretical fracture strength values measured from the monolayer ^{10}BN and ^{11}BN demonstrate that there was no effect of defect on our mechanical measurements.

(6) The bulk $^{\text{Nat}}\text{BN}$ and isotopic BN crystals used for exfoliating atomically thin samples were grown by different methods in different labs, i.e., Ba–BN solvent method under high-pressure in Japan and nickel-chromium solvent method at atmospheric pressure in US, respectively. The $^{\text{Nat}}\text{BN}$ from Japan has been used in numerous studies and contains extremely low density of defects, and the mechanical properties of its monolayer have been systematically studied by us [*Nature Communications* 8, 15815 (2017)]. The monolayer ^{10}BN and ^{11}BN showed differences in the average fracture strength are less than 3% compared to that of the $^{\text{Nat}}\text{BN}$. The small difference measured from one-atom-thin material is unlikely to

be caused by defects, as a mono-vacancy in the effective measuring area could reduce the fracture strength by 20% or more.

Changes: We have made the following changes in the Main text

P.2: This can be attributed to less disorder due to small numbers of atoms, no surface defects due to their 2D nature, simple chemical composition, as well as free of in-plane defects in the effective measuring area that is normally less than 20 nm in radius in atomic force microscope (AFM) based indentation method.

P.5: The ^{10}BN and ^{11}BN bulk crystals were grown by the nickel-chromium solvent method at atmospheric pressure,¹⁶ and the single crystal domains without defects were a few tens of microns¹⁸.

P.5: The transmission electron microscopy (TEM) study of the suspended atomically thin isotopic BN was reported in our previous publication¹⁷.

P.7: Because the single crystal domains without defect were a few tens of microns¹⁸ and our effective measuring radius was less than 20 nm, the chance to encounter defects during the indentation was small.

P.11: Note the measured strength of 2D materials is extremely sensitive to defects, and even one defect in the effective indentation area could cause the pre-mature fracture of atomically thin materials and dramatically decrease their measured fracture strength values³²⁻³⁷.

3. The manuscript may lack a more comprehensive comparison to experimental data, which may limit the impact of this study. While the author briefly mentioned some previous experimental studies about the isotope effect, they do not provide a detailed comparison of their results to these studies. As for simulation results, it could provide important insights into the accuracy of their simulation approach if the authors can compare it with other research, such as some simulation results on graphene.

Response and changes: We mostly agree with the Reviewer and have added the isotope's effect on graphene in the main text. However, due to the mostly contradictory experimental

data from previous studies on the isotope effect on mechanical properties, *e.g.* ^{13}C and $^{\text{Nat}}\text{C}$ diamond [Ref.13 and 14], we stand that it would be better not to directly compare our results with the previous controversial results. On the other hand, our results are consistent with the DFT calculations, providing additional evidence for the effect.

P.16: These anomalous effects could be readily verified on ^{12}C and ^{13}C graphene experimentally, though it has been theoretically predicted by MD simulations that the Young's modulus^{42,43} and fracture strength⁴³ of graphene decreased with the increase of carbon isotope mass, *i.e.* ^{12}C , ^{13}C and ^{14}C .

4. The authors did not fully discuss how detailed description of the computational details of their DFT simulations. This lack of information makes it difficult for readers to assess the accuracy and reliability of the DFT results.

Change: More details on the calculation steps, accuracy and parameters have been added to Method, and the DFT input files and structural files have been added as Supplementary Materials.

P.20: DFT calculations. All first-principles calculations were carried out based on DFT using VASP 5.4.4 code within the PAW-PBE scheme. The generalized gradient approximation along with the DFT-D3 (Grimmer) functional was used, with a well-converged plane-wave cutoff of 800 eV. The atomic coordinates were allowed to relax until the forces on the ions were less than 1×10^{-3} eV \AA^{-1} under the conjugate gradient algorithm. For sufficient accuracy, the electronic convergence was set to 1×10^{-8} eV. The reduced Brillouin zone was sampled with a Γ -centered k-grid meshes of $24 \times 24 \times 1$ and $24 \times 24 \times 6$ for monolayer/bilayer and bulk *h*BN, respectively, which was sufficient within the k-grid mesh density lower than 0.02. A 15 \AA vacuum space was used in all calculations. For different B isotopes, we used different atomic masses, *i.e.* 10.013 a.u. and 10.811 a.u. for ^{10}B and ^{11}B , respectively. The lattice parameters and atomic positions of ^{10}BN and ^{11}BN unit-cells were fully relaxed using the cutoff energy of 800 eV, which was sufficient for the elemental B or N

with the maximum cutoffs of 318 and 400 eV. The force and electronic convergence on each atom were set to be 1×10^{-3} eV \AA^{-1} and 1×10^{-8} eV, respectively. For the monolayer, only the lattice parameters and atomic positions in the two-dimensional plane were relaxed within the fixed symmetry of P-6M2 (No. 183). The DFT-D3 (Grimme) functional was also used, and the Gaussian smearing was utilized with the SIGMA value of 0.01. To calculate the elastic constants, we chose seven compression-tensile structures ranging from -1.5% to 1.5% with an interval of 0.5% compared with the optimized lattice. Phonon spectra were calculated using the VASP 5.4.4 and Phonopy 1.12.2 codes with a supercell of $2 \times 2 \times 1$. To calculate the sliding energy, we chose nine intermediate structures along the periodic direction. To include thermal effects, we calculated the energy of phonons using the quasi-harmonic approximation. Thermal properties such as free energy of phonons, heat capacity, and entropy were computed under temperatures ranging from 0 K to 300 K with compression-tensile structures to confirm the most stable structures with finite temperatures. We further calculated the free energy of compression-tensile structures and obtain the mechanical properties. The corresponding input and structural files are provided (see Supplementary Material).

5. The AFM results indicate the thickness of 1L BN is 0.59 nm, while the authors used the theoretical value (0.334 nm) in their calculation. The author may need to explain why the measured result is much higher than the theoretical value. If the measurement results are within the error bar, then how can the author make sure the sample is mono-layer instead of double-layer?

Response: The theoretical thickness of monolayer BN, i.e. 0.334 nm is equivalent to the distance between two adjacent layers of BN. The theoretical thickness of multilayer BN can be calculated by multiplying the number of layers by the thickness of a monolayer, i.e. $0.334N$ ($N=1,2,3,\dots$) nm. These theoretical values/constants must be used in calculating the mechanical properties. However, the experimental thickness is always larger due to 1) AFM tip convolution; 2) the gap between BN and substrates always larger than between BN and BN layers due to intrinsic reason and sometime the presence of moisture; 3) substrate

roughness; 4) potential surface contaminations (e.g., from air); 5) wrinkles and ripples of 2D materials. The theoretical thickness of bilayer BN is 0.67 nm, so the experimental thickness must be larger than 0.67 nm. Hence the 0.59 nm thickness must be monolayer. Additionally, we have confirmation of thickness from Raman. Furthermore, the thickness of the BN (e.g., 1L vs. 2L) was also determined based on the load-displacement curves from the AFM indentation, as the 2D Young's modulus and fracture load of 2L BN roughly double those of 1L. Moreover, our past experience in studying the mechanical and other properties of several atomically thin materials, such as *Nature Communications* 8, 15815 (2017); *Nature Communications* 9, 1 (2018); *Science Advances* 5, eaav0129 (2019); *Physical Review Letters* 125, 085902 (2020); *ACS Nano* 15, 2600 (2021); *ACS Nano* 8, 1457 (2014); *Nano Letters* 21, 3379 (2021); *Nano Letters* 15, 218 (2015); *Advanced Functional Materials* 26, 8356 (2016) gives us additional confidence on our analysis as the layer number determination is a routine practice in our measurements.

Overall, while the manuscript presents interesting results and is well-written, there are a few major revisions that could be improved to enhance the impact and generalizability of the study.

Response: We thank the Reviewer for his/her comments and questions. We are confident that the updated version of the manuscript addressed all points raised by the Reviewers with the additional data and discussions enhancing the impact and generalisation of our results to other systems. Moreover, the methods and protocols included in our work provide guidelines for exploration of other new and unique mechanical and physical properties of 2D materials and other materials not yet studied in the community. In this aspect, we believe that *Nature Communications* is the right channel to communicate our timely results.

REVIEWER COMMENTS

Reviewer #1 (Remarks to the Author):

Authors addressed all my concerns. My suggestions are listed below.

1. The noise floor Fig. 1 is much larger than 15 pm.
2. According to Fig. S2, the geometry of the suspended BN film might be not able to be figured out from AFM, due to the XY resolution limit in such complicated configuration.
3. The standard deviation should be shown in the main text, as in ref. 1 in supplementary material (Measurement of the Elastic Properties and Intrinsic Strength of Monolayer Graphene. Science 321, 385-388 (2008). With about 5% error (sd), the property change might be not obvious as shown in Fig. e and f.
4. It's difficult to convince that the test result published in 2019 is still representative for current samples in 2023. No evidence from the current sample is shown about defects in the sample, which is key to assume the mechanical property change only derives from the isotope effect.

Reviewer #2 (Remarks to the Author):

The authors have taken adequately the remarks and comments in my review, with fairly explanations. The results sound good and the deductions are well argued. I can recommend the publication of the paper.

Reviewer #3 (Remarks to the Author):

Thank you so much for all your responses and explanation. All my concerns have been addressed in the revisions. Overall, the results are interesting and the manuscript looks more sophisticated after the edit. I will be happy to see the manuscript published.

We thank again the Reviewers. Reviewer #2 and #3 recommended publication, and Reviewer #1's comments are addressed below:

REVIEWER COMMENTS

Reviewer #1 (Remarks to the Author):

Authors addressed all my concerns. My suggestions are listed below.

1. The noise floor Fig. 1 is much larger than 15 pm.

Response: In the previous comment, the Reviewer asked about the noise floor to see the accuracy in determining the layer number of the BN samples. AFM is a routine method to do this, and more importantly, we already clearly explained in the previous response letter that the layer number could also be determined by the mechanical data from indentation:

"On the other hand, the thickness of the BN (e.g., 1 Layer vs. 2L) could also be determined based on the load-displacement curves from the AFM indentation, as the 2D Young's modulus and fracture load of 2L BN double those of 1L."

Anyway, the source data for the AFM image in Fig.1b has been uploaded to Figshare, as suggested by the Editor.

2. According to Fig. S2, the geometry of the suspended BN film might be not able to be figured out from AFM, due to the XY resolution limit in such complicated configuration.

Response: We don't understand why Reviewer thought the geometry of the suspended BN film could not be figured out by AFM and how this (i.e. XY limitation in AFM) has anything to do with the validity of our conclusion.

Anyway, the geometry of the suspended BN film was determined by the pre-fabricated micro-wells in the substrates and could be measured by AFM, as shown in Fig. 1b and S2. We followed the mostly used methodology for the mechanical measurements on 2D materials, including the AFM method to measure the diameter of the suspended 2D materials. From inaccuracy point of view, small changes in the diameter of the suspended film do not affect Young's modulus at all and only has a small impact on fracture strength. More importantly, we used the same substrates and methodology for comparing the isotope effect on the mechanical properties of ^{10}BN , ^{11}BN and $^{\text{Nat}}\text{BN}$ monolayers, so the XY resolution limit in the AFM should not affect the conclusion.

Change: P.8 The radius was determined by AFM, and the inaccuracy had no effect on the Young's modulus and only a negligible effect on fracture strength.

3. The standard deviation should be shown in the main text, as in ref. 1 in supplementary material (Measurement of the Elastic Properties and Intrinsic Strength of Monolayer Graphene. Science 321, 385-388 (2008). With about 5% error (sd), the property change might be not obvious as shown in Fig. e and f.

Response: The standard deviations have been added to the main text:

P.9: The E^{2D} values with standard deviations were 299 ± 22 , 284 ± 21 , and 281 ± 10 N/m for 1L ^{10}BN , $^{\text{Nat}}\text{BN}$, and ^{11}BN , respectively.

P.11: The strength values of the 1L ^{10}BN , $^{\text{Nat}}\text{BN}$, and ^{11}BN with standard deviations were 75.7 ± 7.3 GPa, 73.7 ± 7.4 GPa and 73.1 ± 5.2 GPa, respectively.

4. It's difficult to convince that the test result published in 2019 is still representative for current samples in 2023. No evidence from the current sample is shown about defects in the sample, which is key to assume the mechanical property change only derives from the isotope effect.

Response: We started this project in Apr 2018 (it is very time-consuming to measure the mechanical properties of 2D materials and interpret the data), and the isotopic hBN crystals used in this project were produced by the same method as in the 2019 reference [*Communications Physics* 2, 43 (2019) or Ref.18].

We think adequate explanations have been given on the defects in the previous response letter, e.g. the 6 points regarding defects, especially the 5th and 6th points.

Change: P.5 The 10BN and 11BN bulk crystals were grown by the nickel-chromium solvent method at atmospheric pressure,¹⁶ and the single crystal domains without defects were a few tens of microns, as indicated by a previous study involving crystals prepared by the same method at similar time¹⁸.

Reviewer #2 (Remarks to the Author):

The authors have taken adequately the remarks and comments of my review, with fairly explanations. The results sound good and the deductions are well argued. I can recommend the publication of the paper.

Reviewer #3 (Remarks to the Author):

Thank you so much for all your responses and explanation. All my concerns have been addressed in the revisions. Overall, the results are interesting and the manuscript looks more sophisticated after the edit. I will be happy to see the manuscript published.